# Testicular “Inherited Metabolic Memory” of Ancestral High-Fat Diet Is Associated with Sperm sncRNA Content

**DOI:** 10.3390/biomedicines10040909

**Published:** 2022-04-15

**Authors:** Luís Crisóstomo, Matthieu Bourgery, Luís Rato, João F. Raposo, Rachel L. Batterham, Noora Kotaja, Marco G. Alves

**Affiliations:** 1Departamento de Anatomia, e UMIB—Unidade Multidisciplinar de Investigação em Biomedicina, ICBAS—Instituto de Ciências Biomédicas Abel Salazar, Universidade do Porto, 4050-313 Porto, Portugal; luis.d.m.crisostomo@gmail.com; 2Laboratory for Integrative and Translational Research in Population Health (ITR), University of Porto, 4050-600 Porto, Portugal; 3Integrative Physiology and Pharmacology Unit, Institute of Biomedicine, University of Turku, Kiinamyllynkatu 10, 20520 Turku, Finland; matthieu.bourgery@utu.fi (M.B.); nookot@utu.fi (N.K.); 4Polytechnic Institute of Guarda, School of Health, 6300-035 Guarda, Portugal; luis.pedro.rato@gmail.com; 5Faculty of Health Sciences, University of Beira Interior, Av. Infante D. Henrique, 6200-506 Covilhã, Portugal; 6NOVA Medical School, New University Lisbon, 1250-189 Lisbon, Portugal; filipe.raposo@apdp.pt; 7Associação Protetora dos Diabéticos de Portugal (APDP), 1250-189 Lisbon, Portugal; 8Centre for Obesity Research, Department of Medicine, University College London (UCL), London WC1E 6JF, UK; r.batterham@ucl.ac.uk; 9National Institute for Health Research, Biomedical Research Centre, University College London Hospital (UCLH), London W1T 7DN, UK; 10Biotechnology of Animal and Human Reproduction (TechnoSperm), Institute of Food and Agricultural Technology, University of Girona, 17003 Girona, Spain; 11Unit of Cell Biology, Department of Biology, Faculty of Sciences, University of Girona, 17003 Girona, Spain

**Keywords:** high-fat diet, sperm parameters, sncRNA, paternal epigenetic inheritance, inherited metabolic memory

## Abstract

Excessive adiposity caused by high-fat diets (HFDs) is associated with testicular metabolic and functional abnormalities up to grand-offspring, but the mechanisms of this epigenetic inheritance are unclear. Here we describe an association of sperm small non-coding RNA (sncRNA) with testicular “inherited metabolic memory” of ancestral HFD, using a transgenerational rodent model. Male founders were fed a standard chow for 200 days (CTRL), HFD for 200 days (HFD), or standard chow for 60 days followed by HFD for 140 days (HFDt). The male offspring and grand-offspring were fed standard chow for 200 days. The sncRNA sequencing from epidydimal spermatozoa revealed signatures associated with testicular metabolic plasticity in HFD-exposed mice and in the unexposed progeny. Sperm tRNA-derived RNA (tsRNA) and repeat-derived small RNA (repRNA) content were specially affected by HFDt and in the offspring of HFD and HFDt mice. The grand-offspring of HFD and HFDt mice showed lower sperm counts than CTRL descendants, whereas the sperm miRNA content was affected. Although the causality between sperm sncRNAs content and transgenerational epigenetic inheritance of HFD-related traits remains elusive, our results suggest that sperm sncRNA content is influenced by ancestral exposure to HFD, contributing to the sperm epigenome up to the grand-offspring.

## 1. Introduction

The adoption of high-fat diets (HFDs) associated with increased adiposity, often from early life, is considered one of the major contributors for the present “fat pandemic”. Concurrently, the prevalence of non-communicable diseases, such as type 2 diabetes (T2D), has increased proportionally. The most recent figures by the World Health Organization (WHO) and by the International Diabetes Federation (IDF) estimate that more than 500 million people worldwide have obesity, and 463 million suffer from T2D [1,2]. These reports also highlight the earlier age of onset of obesity-related diseases, raising concerns about its consequences to adults of active and reproductive age. In addition, reports suggest an influence between unhealthy diets, as promoters of overweight and obesity, and health outcomes in progeny, particularly the onset of metabolic disease [3,4]. Therefore, there is a rising concern regarding the long-term consequences of excess adiposity associated with the adoption of HFD to prospective parents and their offspring.

The role of sperm small non-coding RNA (sncRNA) in epigenetic inheritance has been suggested by several authors [5,6]. Recent findings show that tRNA-derived small RNAs (tsRNA), the most abundant sncRNA biotype in sperm, can induce an acquired pathology in non-exposed mice [7], without the influence of any other epigenetic mechanism. Sperm miRNA and tsRNA content have also been shown to be crucial for mammalian embryo development [8]. Besides these functions directly related to epigenetic inheritance, other sncRNA biotypes are crucial for spermatogenesis. For instance, piwi-interacting RNAs (piRNAs) expressed by male germ cells during spermatogenesis have a critical role in silencing transposable elements [9,10].

Obesity, T2D, and diet alter sperm sncRNA content [11,12,13], and those changes may persist in the offspring for several generations [14,15]. Our group previously showed detrimental effects of HFD exposure to mice testicular metabolism and function after feeding with HFD for 200 days from weaning or HFD for 60 days from weaning and then fed with standard chow for 140 days, compared to littermates exclusively fed with standard chow [16,17]. These changes included a higher proportion of immotile, non-viable, and morphologically abnormal spermatozoa; alterations in the testicular content of several metabolites and fatty acids; and differences in the relative testicular proportions of saturated fatty acids (SFAs), monounsaturated fatty acids (MUFAs), ω3 polyunsaturated fatty acids (PUFAs), and ω6 PUFAs [16,17]. We have further observed the inheritance of these changes in testis up to two generations, a phenomenon that we coined as “inherited metabolic memory” [18,19]. Particularly, we observed changes in the testicular content of metabolites involved with the energy-obtaining pathways and insulin resistance, and in the proportion of ω3 PUFAs and ω6 PUFAs in the grand-offspring of mice exposed to HFD, compared to the grand-offspring of mice fed with standard chow [18,19]. However, the mechanisms underlying this heritable metabolic fingerprint of ancestral exposure to HFD were unknown. In light of their response to increased adiposity due to diet and their potential to mediate embryo development, sperm sncRNAs are a potential vehicle of the “inherited metabolic memory” from fathers to sons and grandsons.

In this study, we describe sperm sncRNA response to lifelong exposure to HFD and exposure to HFD until early adulthood. We further investigate if ancestral exposure to HFD is reflected in the sperm sncRNA profile in the offspring. This work was conducted by using the same animal model as our recent studies [16,17,18,19], expanding our previous findings and providing the mechanistic evidence of “inherited metabolic memory” in the testis of mice.

## 2. Materials and Methods

### 2.1. Animal Model

We designed a transgenerational ancestral exposure model based on three generations of C57BL6/J mice [18,19]. Normoponderal male and female mice, fed with a standard chow (#F4031, BioServ, Flemingtown, NJ, USA—Carbohydrate: 61.6%, Protein: 20.5%, Fat: 7.2–16.3% Kcals) and water ad libitum, were used to produce the exposed founders (Generation F0). After weaning (21–23 days), F0 mice (*n* = 36) were randomly divided in three groups: control (CTRL) (*n* = 12), HFD (*n* = 12), and HFDt (*n* = 12). Each group was assigned to a diet regimen: CTRL—standard chow (#F4031, BioServ, Flemingtown, NJ, USA) during 200 days; HFD—fat-enriched diet (#F3282, BioServ, Flemingtown, NJ, USA—carbohydrate = 35.7%, protein = 20.5%, and fat = 36.0–59.0% Kcals) during 200 days; and HFDt—fat-enriched diet (#F3282, BioServ, Flemingtown, NJ, USA) during 60 days, then switched to standard chow (#F4031, BioServ, Flemingtown, NJ, USA) during 140 days. When animals reached 120 days post-weaning age, F0 mice were mated with normoponderal, chow-fed, same-age, randomly selected females to generate the unexposed offspring (Generation F1). Mating was performed in mating pairs, without access to water and food for 6 h a day. The process was repeated over 8 days. After weaning, the F1 mice were assigned to the same experimental group as their fathers: CTRL_F1—Offspring of CTRL (*n* = 12), HFD_F1—Offspring of HFD (*n* = 12), and HFDt_F1—Offspring of HFDt (*n* = 12), but all mice were fed with standard chow (#F4031, BioServ, Flemingtown, NJ, USA) during 200 days. F1 mice were mated in the same scheme as their progenitors to generate the unexposed grand-offspring (Generation F2). F2 mice were assigned to experimental groups in the same way as their fathers (12 animals per group), and fed with standard chow for 200 days after weaning. Food and water were supplied without restrictions to all generations. Mice from all generations were killed by cervical dislocation 200 days after weaning, and tissues were collected for further analysis. Total body weight, water, and food intake were monitored weekly, from weaning to sacrifice. The animal model is compliant with the ARRIVE guidelines, was internally reviewed by the Organization for Animal Welfare (ORBEA) and approved and licensed by the Portuguese Veterinarian and Food Department (DGAV) with the registration number 0421/000/000/2016. All animal experiments were conducted in the Animal Facility of the Research Center in Health Sciences (CICS) of the University of Beira Interior (Covilhã, Portugal), according to the “Guide for the Care and Use of Laboratory Animals” published by the US National Institutes of Health (NIH Publication No. 85–23, revised 1996) and European rules for the care and handling of laboratory animals (Directive 86/609/EEC). None of the interventions performed on the animals required the use of anesthetics.

### 2.2. Sperm Collection and RNA Extraction

Epididymides were isolated and placed in pre-warmed (37 °C) Hank’s Balanced Salt Solution (HBSS), pH 7.4. Then small incisions were made using a scalpel blade, and the suspension was incubated for 5 min (37 °C) to allow spermatozoa to swim out of the epididymis. The purity of the sperm solution was accessed in sperm swabs stained by eosin–nigrosin and observed by an optical microscope (Appendix A). The solution was then snap-frozen in liquid nitrogen and stored at −80 °C. This solution was later thawed, and total RNA was extracted based on the acid guanidinium thiocyanate–phenol–chloroform extraction method [20]. Before adding 1 mL of the ready-to-use reagent for total RNA isolation (EXTRAzol, Blirt S.A., Gdańsk, Poland), the sperm solution was centrifuged at 200× *g* for 5 min at room temperature, and the supernatant was discarded, according to the manufacturer’s instructions (protocol for cells grown in suspension). RNA pellets were resuspended in 20 μL of DEPC-treated water and stored at −80 °C until library preparation.

### 2.3. cDNA Library Preparation and Next Generation Sequencing

Before library preparation, RNA concentration and integrity were screened by using an Agilent 2100 Bioanalyzer (Agilent Technologies, Santa Clara, CA, USA). The RNA integrity number (RIN) was globally low (ranging 2–5), a normal value for RNA obtained from mature spermatozoa, which lacks 18S and 28S rRNA peaks [21]. RNA concentration was considered acceptable between 1 ng and 2 μg. Library preparation was performed with NEXTFLEX^®^ Small RNA-Seq Kit v3 for Illumina (Bioo Scientific Corporation, Austin, TX, USA) according to the manufacturer instructions. Pooled libraries were sequenced by using Next-Generation Sequencing (NGS), in single-end mode, on NextSeq 500 with NextSeq 500/550 High Output Kit version 2, 75 cycles (Illumina, San Diego, CA, USA). All pooled libraries passed Illumina’s default quality control.

### 2.4. Processing and Annotation of Sequenced Reads

Cutadapt version 2.10 [22] was used to trim the 3′ adapter (TGGAATTCTCGGGTGCCAAGG) and four nucleotides upstream and downstream the adapter from sequenced reads. The adaptor is a nucleotide sequence added during library preparation to allow the amplification of sncRNAs. This sequence will also be sequenced as part of the fragments, therefore contaminating them and hindering the correct annotation of the RNA sequences. Only trimmed reads containing adaptor sequence, and with 80% of the nucleotides having Illumina quality scores (Q-scores) > 20, were retained. Trimmed reads were mapped to small RNAs by using SPORTS 1.1 (https://github.com/junchaoshi/sports1.1; accessed on 1 June 2020), a perl-based analytical workflow tweaked for the annotation of sncRNAs [23]. SPORTS does not presently annotate piwi-interacting RNAs (piRNA), but it can identify sequences as piRNA. Representative sperm RNA diagnosis statistics obtained by SPORTS 1.1 are represented in Appendix A. We used the default settings and custom databases for the mouse genome: full genome from GRCm38/mm10 UCSC (https://hgdownload.soe.ucsc.edu/downloads.html#mouse; accessed on 1 June 2020), miRNA from miRbase 22 (http://www.mirbase.org/; accessed on 1 June 2020), rRNA-derived small RNA (rsRNA) from NCBI/Nucleotide (available at SPORTS repository), tRNA-derived small RNA (tsRNA) from GtRNAdb (http://gtrnadb.ucsc.edu/genomes/eukaryota/Mmusc10/; accessed on 1 June 2020), and piRNA from pirBase 2.0 (http://www.regulatoryrna.org/database/piRNA/download.html; accessed on 1 June 2020). Fragment counts for specific sequences and for sncRNA biotypes (miRNA, rsRNA, tsRNA, and piRNA), per sample, were based on the default annotations in SPORTS result output files.

### 2.5. Annotation of piRNA Clusters and Repeats

SPORTS 1.1 is unable to annotate piRNAs and repRNAs (transcripts from transposable elements, such as LINEs and SINEs, among others); therefore, an alternative annotation using HISAT2 [24], SAMtools [25], and featureCounts [26] was applied. The annotation of individual piRNA sequences is not consensual, due to their ambiguity and multiplicity of targets; thus, a reliable alternative is the annotation of sequences according to the pachytene piRNA cluster from which they are transcribed [27,28]. Trimmed samples were annotated against the total mouse genome from GRCm38/mm10 UCSC database, using HISAT2. The output files containing the fragment reads were then indexed by using SAMtools. Eventually, index files were used to map piRNAs to their precursor cluster in the mouse piRNA cluster database [28] (https://www.smallrnagroup.uni-mainz.de/piCdb/; accessed on 3 June 2020), and interspersed repeats to their genomic region in to Dfam 3.1 database [29] (https://www.dfam.org/home; accessed on 3 June 2020), using featureCounts to aggregate the final sequence counts.

### 2.6. Term Enrichment Analysis

Differently expressed sncRNAs were selected for term enrichment analysis based on gene ontology (GO) terms [30,31]. First, miRNA and tsRNA targets were estimated by using sRNAtools (https://bioinformatics.caf.ac.cn/sRNAtools; accessed on 15 August 2020) [32], based on the union of the potential gene targets calculated by miRanda [33] and RNAhybrid [34] interaction algorithms. The miRanda parameters were set to minimal free energy cutoff = −20 and miRanda score cutoff = 160, whereas RNAhybrid parameters were set to energy cutoff = −20 and *p*-value cutoff = 0.01. Differently expressed piRNA clusters were added to the list of potential miRNA and tsRNA targets. This list was uploaded to R 4.1.0 [35] and annotated to gene ontology (GO) terms by using the packages topGO 2.42.0 [36] and GO.db 3.8.2 [Mus Musculus] [37]. Target genes were coalesced, and unique targets were filtered out. GO terms were annotated as for biological process, molecular function, and cellular component. Terms with less than 10 gene annotations were excluded. Term enrichment analysis was performed by using the “weight01” algorithm included in topGO.

### 2.7. Statistics

Raw sequence counts obtained by SPORTS and featureCounts were imported as dataframes into R 4.1.0 [35]. A non-conservative coverage filter was applied to discarded fragments with <17 nucleotides and <0.01 reads per million (RPM) per group. All statistics were performed by using DESeq2 package v1.28.1 [38]. Samples were normalized by the package’s code, based on a Generalized Linear Model (GLM) and Bayesian shrinkage [38]. The differential expression analysis is based on the Negative Binomial distribution, using Wald statistics. Multiple hypotheses are corrected for the False Discovery Rate (FDR) based on the Benjamini–Hochberg method. The proportion of sncRNA families was considered changed if adjusted *p* < 0.1. A fragment was considered differently expressed whenever log2 Fold Change (FC) > |1.5| and adjusted *p* < 0.1. GO term enrichment analysis was tested by Fisher Exact test. GO term annotations were sorted according to their “weight01” score. GO terms were considered significant when *p* < 0.1, regardless of the statistical test. Whenever more than 10 GO terms met the criteria, only the top 10 terms were considered.

## 3. Results

### 3.1. Dietary Changes Affect Sperm sncRNA Content

To understand how the sperm sncRNA profile responds to chronic (200 days) exposure to high-fat initiated at weaning, we collected mature spermatozoa from chow-fed (CTRL) and HFD-fed mice (HFD) for small RNA-sequencing. We also analyzed sperm from mice on transient HFD (HFDt, 60 days of HFD started at weaning, followed by 140 days of chow before sacrifice) to study if the exposure to HFD and adiposity cause irreversible changes in sperm sncRNA content. To get a general view on the mapping of sperm RNA reads into different RNA biotypes, annotated RNA sequences were categorized as miRNA, tRNA-derived small RNA (tsRNA), mitochondrial tRNA-derived small RNA (mtRNA), Piwi-interacting RNA (piRNA), rRNA-derived small RNA (rsRNA), repeat-derived small RNA (repRNA), and Y RNA, and their total counts were normalized by using DESeq2 based on total genome-matched sequences. We compared only animals which have lived during the same period in order to limit the influence of uncontrollable variables that may affect sperm sncRNA content, such as seasonal and environmental variation. This analytical strategy has been commonly adopted in transgenerational inheritance models [39,40].

HFD or HFDt did not cause major changes in the relative fraction of the different RNA biotypes in sperm in any of the generations (F0–F2). Only rsRNA reads were slightly increased in F0 sperm after transient HFDt compared to CTRL (1.16 log2 FC, *p* = 0.0013) and compared to HFD (0.76 log2 FC, *p* = 0.0945) (Figure 1A). Contrary to their F0 progenitors, the F1 offspring of HFDt mice displayed reduced levels of rsRNA reads in sperm than the offspring of CTRL mice (Figure 1A,B) (−1.18 log2 FC, *p* = 0.0128). The size distribution of small RNA reads was as expected (Appendix A) and was not affected by the different diets.

Next, we performed differential expression (DE) analysis between groups to identify any changes in the levels of individual RNAs. We analyzed separated miRNAs and two distinct groups of tsRNAs: transcription initiation RNAs (tiRNAs), also known as tRNA halves, and tRFs (tRNA-derived fragments). The tiRNAs are 29–50 nucleotide sequences resulting from the cleavage of mature tRNAs at the anticodon loop by angiogenin, thus producing a 3′-tRNA half and a 5′-tRNA half. The tiRNAs are involved in the post-transcriptional gene expression regulation via base complementarity to transcription start sites, to RNAse II recognition patterns, and to the ribosome [41,42,43]. Notably, tiRNA expression is triggered in response to oxidative stress to inhibit protein translation and assemble stress granules [41,42]. The tRFs are shorter (16–28) nucleotide sequences resulting from the cleavage of mature tRNA or pre-tRNA by angiogenin and DICER. These sequences are often classified according to the mature tRNA loop they derive from. The tRFs are associated with the silencing of transposable elements [44,45], but their expression is not restricted to male germline cells. We also mapped sncRNA reads to the 214 genomic clusters known to produce piRNAs during spermatogenesis [27,28], as well as interspersed repeat sequences, including transposable elements [27,46] to study if the piRNA clusters and repRNAs are affected. Earlier reports have shown that an HFD affects sperm sncRNA levels, particularly tsRNAs, but also miRNAs [12,47]. Interestingly, in our experimental setup in which HFD was initiated already at puberty, we did not see any changes in sperm RNA levels in HFD vs. CTRL mice (Figure 2A,C). In contrast, HFDt induced changes in tsRNAs and repRNA levels compared to CTRL. All the differently expressed tsRNA sequences (eight up and two down) were tiRNAs, nine of which are encoded by the mitochondrial genome (Figure 2B). Regarding repRNAs, seven sequences were upregulated and two were downregulated (Figure 2D): two Long Interspersed Nuclear Elements (LINEs), three spliceosomal small nuclear RNAs (snRNAs), one Short Interspersed Nuclear Element (SINE), one Signal Recognition Particle RNA (sprRNA), and two simple repeats. Our results show that changes in the diet (from chow to HFD and back to chow) induce more changes in sperm RNA profile than constant long-term HFD starting at weaning, suggesting that mice and their sperm RNA profile can adapt to the chronic HFD.

### 3.2. Dietary Changes Affect Sperm sncRNA Content of the Offspring of Exposed Mice

To study the intergenerational effect of exposure to HFD in sperm sncRNA content, mice founders (Generation F0) were randomly assigned to a normoponderal female in mating pairs 120 days after weaning. The resulting male offspring (Generation F1) were exclusively fed with chow diet for 200 days, independently of the diet regimen of its male progenitor. Littermates were assigned to the same experimental group (lineage) as its progenitor.

The DE analysis revealed that ancestral HFD and HFDt affect sperm RNA levels in the offspring. Interestingly, different biotypes were affected in HFD and HFDt F1 offspring compared to offspring of CTRL group (Figure 3A–I). HFD F1 offspring sperm had drastically lower levels of two miRNAs (215-5p, log2FC = −23.96, *p* < 0.0001 and 3068-5p, log2FC = −6.78, *p* = 0.0701) and one tiRNA (mt_Gly-TCC_3_end, log2FC = −8.65, *p* = 0.046) compared to the CTRL F1 offspring (Figure 3A,D,G). In contrast, HFDt F1 offspring tRF levels were affected with five upregulated and eight downregulated tRFs compared to the CTRL F1 offspring (Figure 3B,E,H).

Interestingly, the DE analysis of piRNA clusters revealed, altogether, 18 clusters that were downregulated in HFD F1 offspring sperm compared to CTRL offspring (Figure 4A). Only one of them was downregulated in HFDt F1 offspring sperm (Figure 4B). This suggests that, although the father’s exposure to chronic HFD does not affect sperm piRNA levels in the exposed animal, it does induce intergenerational changes in F1 offspring spermatogenesis that are reflected as differential levels of piRNAs in the offspring sperm. In addition to piRNAs, a large number of differently expressed sperm repRNAs were found between the HFD F1 offspring and the CTRL F1 offspring (29 up and 51 down) (Figure 4D). Twenty-one sperm repRNAs were also more abundant in the offspring of HFDt compared to the offspring of CTRL (Figure 4E). The different levels of repRNAs in F1 offspring sperm may also originate from the father’s HFD-induced changes in offspring spermatogenesis that affect the expression of repeat-derived RNAs.

### 3.3. Ancestral Dietary Changes Can Be Detected in Sperm of the Grand-Offspring

To study the transgenerational effect of ancestral exposure to HFD in sperm sncRNA content, the mice of Generation F1 were mated in the same conditions as their progenitors. The resulting male offspring were exclusively fed with a chow diet for 200 days and constitute the grand-offspring (Generation F2) of the male founders.

An ancestral chronic HFD did not leave any detectable RNA signatures in F2 sperm, since no differentially expressed RNAs were identified (Figure 5A). However, we identified some changes in the F2 offspring of mice exposed to transient HFDt compared to CTRL. A majority of sncRNAs were not affected, but two miRNAs were upregulated in HFDt F2 offspring sperm compared to CTRL F2 offspring sperm (196a-5p and 196b-5p) and three were downregulated (144-3p, 34c-5p, and 471-5p) (Figure 5B). Moreover, 196a-5p and 196b-5p were also upregulated when the HFDt F2 offspring were compared to the HFD F2 offspring (Figure 5C). In addition, we identified two other upregulated miRNAs (127-3p and 145a-5p) and two downregulated (135b-5p and 32-5p) miRNAs between these groups (Figure 5C), possibly contributing to the testicular and sperm phenotype differences that were reported earlier between these groups [18,19].

### 3.4. Targets of Differently Expressed Sperm sncRNAs Are Related with Crucial Biological Processes for Testicular Function

To estimate the potential role of the sncRNA sequences in testicular function and sperm parameters, targets of differently expressed miRNAs and tsRNAs were predicted by using bioinformatic tools. These targets and the differently expressed piRNA clusters were then annotated according to gene ontology (GO) terms for biological processes (Figure 6), molecular function (Appendix A), and cellular component (Appendix A), and their overall GO term impact was calculated by using the R package topGO. Briefly, the GO term impact was evaluated as a function of the pool of entered genes, the number of genes associated with a term, and the number of “hits” between the entered gene list and the term gene list.

The most significant GO terms resulting from the targets of the differently expressed sncRNAs between HFDt and CTRL mice are the sensory perception of smell and the G-protein-coupled receptor signaling pathway (Figure 6), indicating a possible impact in the chemotaxis of mature spermatozoa toward the oocyte.

Interestingly, this term is also prominent, considering the differences between the offspring of HFDt compared to the offspring of HFD. Compared to the offspring of CTRL, sperm sncRNAs from the offspring of HFD have several changes in GO terms of biological processes linked to cell motility, cytoplasm projections, and even sperm activation (cell morphogenesis, establishment or maintenance of cell polarity, locomotory behavior, and positive regulation of transmembrane receptor protein serine/threonine kinase pathway). Comparing the potential impact of differently expressed sperm sncRNAs between the offspring of HFD and the offspring of HFDt, the most relevant biologic process terms are related to antioxidant response and cell proliferation, especially of bone-related cells. However, in this comparison, relevant molecular function terms are related to ATP binding and energy-sensing receptors (Appendix A), whereas relevant cellular component terms are related to the cytoskeleton and lipid membranes (Appendix A). Taken together, these results suggest an impact in the plasticity of the Sertoli cell membrane that is crucial for spermatogenesis, but also for sperm morphology and motility.

The targets of the differently expressed sperm sncRNAs between the grand-offspring of HFDt and grand-offspring of CTRL mice are mostly associated with metabolic regulation, cell adhesion, and cell differentiation (Figure 6). The estimated impacts of the differently expressed sperm sncRNA of the grand-offspring of HFD compared to the grand-offspring of HFDt are linked to apoptosis, lipid metabolism, chemotaxis, and inflammation (Figure 6). These findings suggest that the maintenance of the blood–testis barrier is affected by ancestral excess adiposity caused by HFD, while the reversion to standard chow promotes changes in testicular metabolism, reflected even in the grand-offspring.

## 4. Discussion

The increasing prevalence of obesity and associated comorbidities is primarily associated with risk behaviors, such as excessive caloric intake and physical inactivity. However, several studies have suggested a modulation of the predisposition to weight gain and the onset of non-communicable diseases due to ancestral exposure HFD [15,48,49], even in humans [3,4]. We have previously shown that ancestral exposure to HFD causes metabolic and functional changes in testes up to two generations, a phenomenon we have coined as “inherited metabolic memory” [18]. In this work, we studied the role of sperm sncRNA content in the transmission of acquired testicular metabolic adaptions to HFD to the offspring and grand-offspring of mice exposed to HFD.

In previous works [16,17], we have shown in mice that the adoption of HFD from weaning led to an obese phenotype with markers of pre-diabetes and marked metabolic reprogramming of the testicular tissue associated with lower sperm quality. We have further reported that the reversion of the HFD during early adulthood prevented the pre-diabetic phenotype but could not revert the testicular metabolic reprogramming and rescue sperm quality. Despite that, the replacement of the HFD with a chow diet promoted changes in testicular metabolism, notably the fatty acid metabolism and fatty lipid composition of the testes. Curiously, in this work, we exclusively found changes between the sperm sncRNA content of transient HFD mice (HFDt) and controls (CTRL) (Figure 4 and Figure 7A). A majority of differently expressed sncRNA sequences in sperm are tiRNAs of mitochondrial origin (Table 1). This finding supports our previous hypothesis that the reversion of diet promotes the mobilization of fatty acids in testes that are used by mitochondria as energy substrate [17]. Moreover, as previously reported [16,50], testicular metabolism is susceptible to environmental factors, such as diet before sexual maturation. Hence, the changes observed in sperm sncRNA content of HFDt compared to CTRL mice may reflect the limited metabolic reprogramming in response to a new environmental factor (chow diet). In addition, tiRNAs may also influence the post-transcriptional regulation of gene transcripts via base-pair complementarity with gene transcription start sites and RNA polymerase II binding motifs [43], and may inhibit gene translation via the assembly of stress granules [41,42]. We have estimated the potential targets of differently expressed tiRNAs and annotated then according to GO terms to predict their biological role (Figure 6). This analysis estimated a significant impact in GO terms related to sensory perception of smell. Odorant receptors have been identified in sperm for several years and have been implicated in sperm capacitation, acrosome reaction, and oocyte fertilization [51,52]. However, as mature spermatozoa are transcriptionally silent, it is difficult to associate the differently expressed tiRNAs with decreased expression of odorant receptors in sperm. Moreover, we have not observed differences in fertility rates among founder mice [18].

Another striking difference between the results of this study and our previous studies is related to the offspring of the mice exposed to HFD [18,19]. The intergenerational effects of HFD and HFDt had a limited impact in the phenotype of the mice offspring, even on testicular metabolism. Comparing the testicular metabolic fingerprints of the offspring, it was not possible to segregate samples according to the diet of the founders. In contrast, in this study, we have observed major differences in sperm sncRNA content according to the diet of the founder mice (Figure 4, Figure 5 and Figure 7A). Both the offspring of HFD and the offspring of HFDt had a large number of differently expressed sperm repRNAs compared to the offspring of CTRL (Figure 4D,E). Notably, all the differently expressed sperm repRNAs between HFDt and CTRL were upregulated. The expression of genomic transposable elements must be repressed throughout meiosis to secure the integrity of the DNA. PiRNAs have a central role in this function and are required to achieve a normal spermatogenesis [9,53,54]. Moreover, tRFs have been suggested to play a similar role [44,45], but, contrary to piRNAs, their expression is not restricted to male germline cells. Curiously, we have found differently expressed sperm piRNAs in the offspring of HFD compared to CTRL (Figure 4A), and differently expressed tRFs in the offspring of HFDt compared to CTRL (Figure 3H). Therefore, our data suggest that control over the transcription of transposable elements is destabilized by paternal HFD via the piRNA-directed cleavage of transcripts in the chromatoid body of spermatocytes and spermatids [10], and by paternal HFDt via Angiogenin/DICER cleavage of mature tRNAs of germline cells [55].

An interesting finding from our previous transgenerational studies [18,19] was the reappearance of the defective sperm phenotype, characterized by decreased sperm concentration, in the grand-offspring of the mice fed with HFD or HFDt [18]. Moreover, the testicular metabolic profile had clear differences according to the ancestral exposure to HFD. In this study, and similarly to mice founders, the grand-offspring of mice fed with HFDt had a limited number of differently expressed sperm sncRNA compared to CTRL mice, and the grand-offspring of HFD had no changes compared to CTRL (Figure 5). Moreover, all the differently expressed sperm sncRNAs in this generation are miRNAs. Interestingly, the disruption of miRNA in mouse germline due to conditional knockout of DROSHA, an enzyme involved in miRNA processing, decreased sperm counts [56]. In mammals, miRNAs are involved in the post-transcriptional regulation of mRNA, inhibiting the translation of gene transcripts by base-pair complementarity binding [57]. The biological impact can therefore be estimated by the complementary to gene transcripts—targets. Due to this mechanism, sperm miRNAs have been particularly associated with embryonic development, although the size of this effect in mammals is controversial. Moreover, sperm miRNA content is influenced by the transcription during spermatogenesis. Similarly to previous studies [12], we have estimated the potential targets of the differently expressed miRNAs to evaluate the potential biological impact according to GO annotation (Figure 6). Despite the limited number of differently expressed miRNAs, their targets are related to biological processes involved in spermatogenesis (regulation of cell differentiation), blood–testis barrier (cell–cell adhesion) and lipid metabolism (regulation of fatty acid metabolic process and phospholipid metabolic process). Nevertheless, the results of this analysis must be discussed critically, as we cannot predict the effects of the differently expressed miRNAs in the grand-grand-offspring, nor prove the expression of miRNA targets in the cell precursors of the collected epidydimal spermatozoa.

With this project we have demonstrated the conditional inheritability of phenotypes related to ancestral exposure to HFD (Figure 7B). The exposure to HFD and the reversion from HFD to chow are the stimuli driving phenotypic changes and, therefore, epigenetic remodeling in somatic cells of the founder mice (Generation F0). This remodeling will also impact the sncRNA content of mature spermatozoa that will originate the offspring. Sperm of the HFD-exposed founder will likely carry other epigenetic signatures of HFD, such as changes in DNA methylation pattern, in protamine retention, and in histone modification [5,6], that will contribute to the phenotype of the offspring from early embryonic development [7,8]. In this study we observed changes in sperm tiRNA and repRNA content, although those changes were unable to induce an evident phenotype in the offspring [18,19]. However, this stimulus has produced a response, likely in the epigenome of somatic cell, that caused several changes in sperm RNA content. Both the offspring of HFD and the offspring of HFDt have differences in repRNA sequences, eliciting an impact of paternal HFD in the ability to silence transcripts from transposable elements of DNA. This inability may, by its turn, be a result of upstream dysfunction of the repeat-silencing mechanisms, namely piRNAs and tRFs. This sperm sncRNA signature, in addition to other epigenetic features carried by the offspring sperm, will be the first environmental stimulus of the grand-offspring of mice founders. This stimulus was able to induce a phenotype in sperm parameters and testicular metabolism [18,19]. As a result, sperm of grand-offspring carry another unique sncRNA signature that will provide the first epigenetic stimulus of the following generation.

## 5. Conclusions

In summary, we have shown that HFD causes a “rippling effect” in HFD-related phenotypes of testicular function and sperm sncRNA content. The “inherited metabolic memory” of ancestral exposure HFD results from a complex interaction between the inherited epigenome and stochastic events that creates a new epigenetic landscape that may not be associated with an evident phenotype, yet produces a new sperm epigenetic signature which will influence the epigenome of the following generation (Figure 7B). However, to support this inheritance mechanism, future studies must investigate other epigenetic factors carried by sperm in response to HFD and HFD-related adiposity, as well as the impact of HFD and adiposity on the epigenome of germ cells and testicular cells, notably Sertoli cells. Moreover, the impact of altered sperm sncRNA content in embryo development could be further studied, for instance, by injecting sperm sncRNAs isolated from HFD-exposed individuals into naïve zygotes [7]. Additionally, qRT-PCR could be used to validate our sncRNA candidates, increasing the number of biological replicates. Although NGS and qRT-PCR are distinct techniques, especially regarding normalization and statistical analyses, it has been demonstrated that the results are comparable, even for miRNA expression [58]. Despite these limitations, we described the complexity of the epigenetic inheritance mechanisms in mammals, and we showed the relevance of sperm sncRNAs to sperm parameters and testicular metabolism. For instance, transcripts from repeating elements have been implicated with success rates of Intracytoplasmatic Sperm Injection (ICSI) [59]; thus, knowing the family history of metabolic disease may help the decision-making process of the best suited Assisted Reproduction Technique (ART) of the infertile couple. However, further studies are needed to translate our findings to human health. Now, we suggest that sperm sncRNAs are a potentially meaningful marker of metabolic and functional changes in testes due to ancestral exposure to HFD. Acknowledging these detrimental effects opens new therapeutical opportunities in idiopathic male infertility to improve the efficiency of ARTs and to curb the perpetuation of acquired metabolic disease.

## Figures and Tables

**Figure 1 biomedicines-10-00909-f001:**
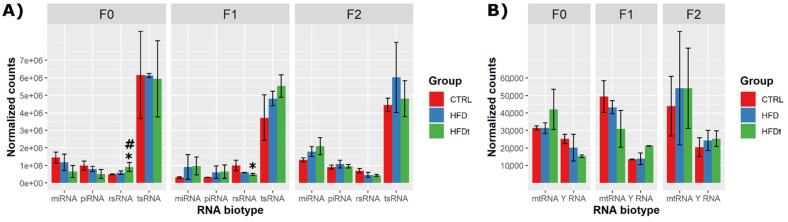
Distribution of sncRNA reads between different RNA biotypes across groups and generations. Results are expressed as the mean of normalized counts ± standard deviation. Generation F0 and F2: *n* = 3 per group. Generation F1: *n* = 2 per group. (**A**) Most representative sperm sncRNA biotypes, and (**B**) mtRNA and Y RNA. Data were tested by Wald’s test, corrected by the Benjamini–Hochberg method, using DESeq2. Significance was considered when *p* < 0.1. * vs. CTRL; # vs. HFD.

**Figure 2 biomedicines-10-00909-f002:**
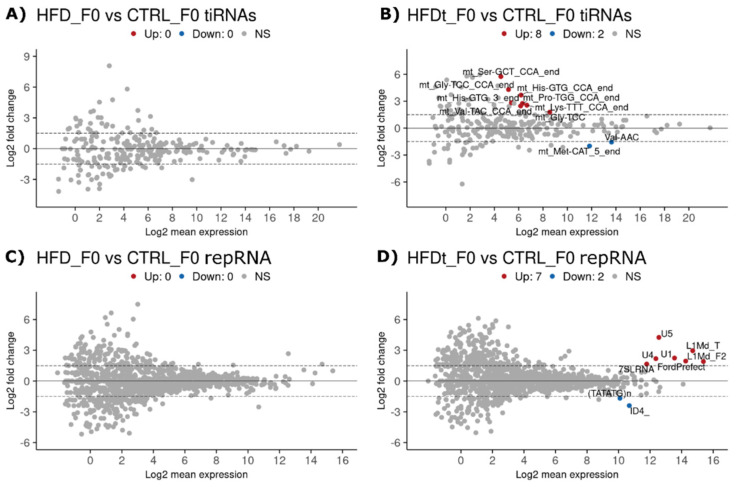
Differently expressed sperm sncRNAs in Generation F0. Sperm sncRNA levels were compared between mice (Generation F0, *n* = 3 per group) fed with standard diet (CTRL), high-fat diet (HFD), or HFD for 60 days that was then replaced by a standard diet (HFDt). Results are presented as MA plots (mean log2 FC vs. log2 mean expression). Differently expressed sequences are highlighted and annotated. Data were tested by Wald’s test, corrected by the Benjamini-Hochberg method, using DESeq2. Significance was considered when *p* < 0.1. (**A**) Transcription initiation RNA (tiRNA), HFD vs. CTRL; (**B**) tiRNA, HFDt vs. CTRL; (**C**) repeat-derived small RNA (repRNA), HFD vs. CTRL; (**D**) repRNA, HFDt vs. CTRL.

**Figure 3 biomedicines-10-00909-f003:**
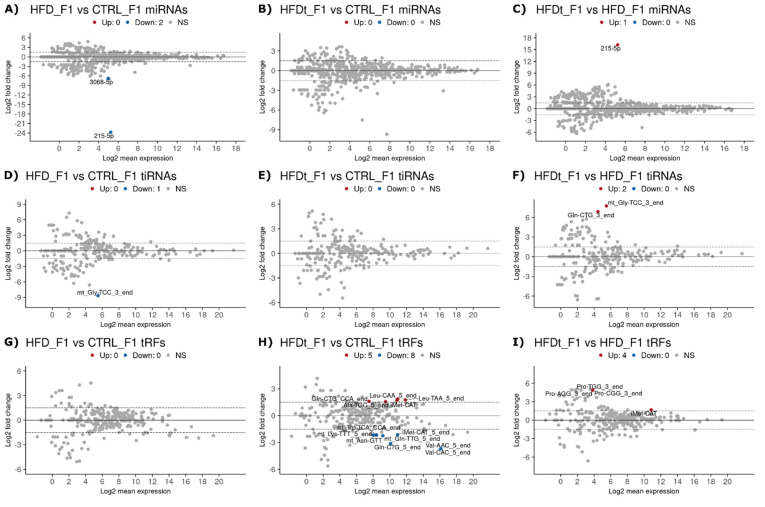
Differently expressed sperm miRNAs, transcription initiation RNA (tiRNA), and tRFs (tRNA-derived fragments) in F1 generation. Sperm sncRNA levels were compared between the offspring (Generation F1, *n* = 2 per group) of mice fed with standard diet (CTRL), high-fat diet (HFD), or HFD for 60 days that was then replaced by a standard diet (HFDt). Results are presented as MA plots (mean log2 FC vs. log2 mean expression). Differently expressed sequences are highlighted and annotated. Data were tested by Wald’s test, corrected by the Benjamini–Hochberg method, using DESeq2. Significance was considered when *p* < 0.1. (**A**) MiRNA, HFD vs. CTRL; (**B**) miRNA, HFDt vs. CTRL; (**C**) miRNA, HFDt vs. HFD; (**D**) tiRNA, HFD vs. CTRL; (**E**) tiRNA, HFDt vs. CTRL; (**F**) tiRNA, HFDt vs. HFD; (**G**) tRF, HFD vs. CTRL; (**H**) tRF, HFDt vs. CTRL; (**I**) tRF, HFDt vs. HFD.

**Figure 4 biomedicines-10-00909-f004:**
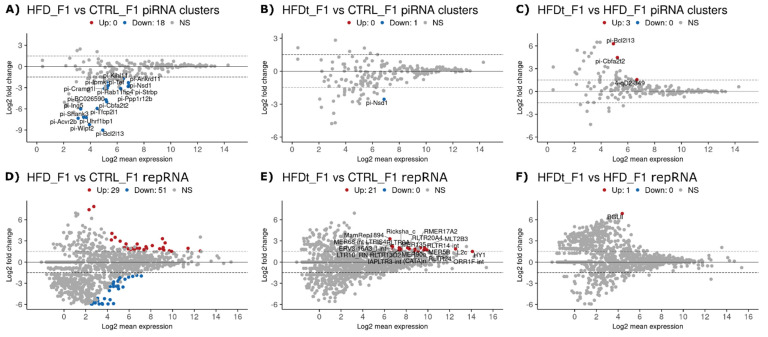
Differently expressed sperm piwi-interacting RNA (piRNA) and repeat-derived RNA (repRNA) sequences between the offspring (Generation F1, *n* = 2 per group) of mice fed with standard diet (CTRL), high-fat diet (HFD), or HFD for 60 days that was then replaced by a standard diet (HFDt). Results are expressed as MA plots (mean log2 FC vs. log2 mean expression). Differently expressed sequences are highlighted and annotated. Data were tested by Wald’s test, corrected by the Benjamini–Hochberg method, using DESeq2. Significance was considered when *p* < 0.1. (**A**) PiRNA, HFD vs. CTRL; (**B**) piRNA, HFDt vs. CTRL; (**C**) piRNA, HFDt vs. HFD; (**D**) repRNA, HFD vs. CTRL; (**E**) repRNA, HFDt vs. CTRL; (**F**) repRNA, HFDt vs. HFD.

**Figure 5 biomedicines-10-00909-f005:**
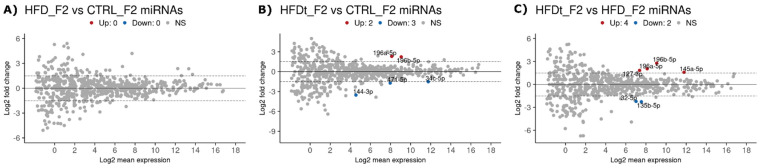
Differently expressed sperm sncRNA sequences between the grand-offspring (Generation F2, *n* = 3 per group) of mice fed with standard diet (CTRL), high-fat diet (HFD), or HFD for 60 days that was then replaced by a standard diet (HFDt). Results are expressed as MA plots (mean log2 FC vs. log2 mean expression). Differently expressed sequences are highlighted and annotated. Data were tested by Wald’s test, corrected by the Benjamini–Hochberg method, using DESeq2. Significance was considered when *p* < 0.1. (**A**) MiRNA, HFD vs. CTRL; (**B**) miRNA, HFDt vs. CTRL; (**C**) miRNA, HFDt vs. HFD.

**Figure 6 biomedicines-10-00909-f006:**
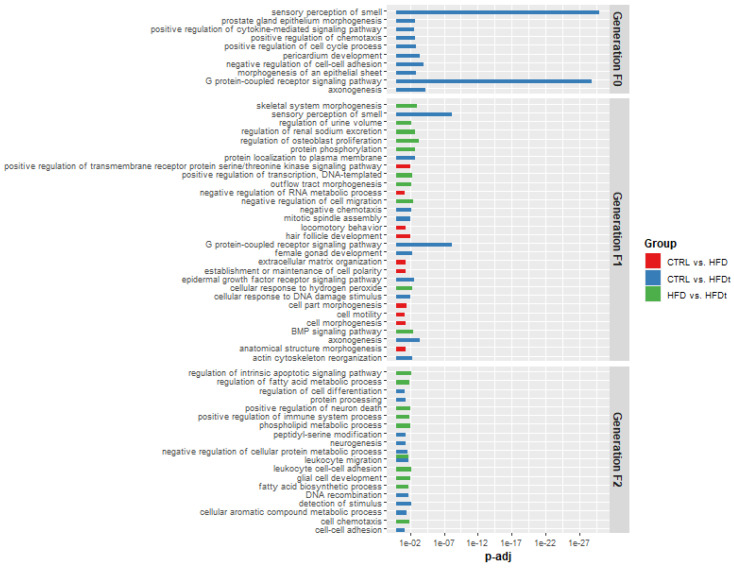
Gene ontology of the targets of differently expressed sperm sncRNAs, according to biological process. Targets of sncRNAs were estimated recurring to the online suite sRNAtools (https://bioinformatics.caf.ac.cn/sRNAtools/; accessed on 15 August 2020). The output was then used for functional annotation based on the Gene Ontology Resource (http://geneontology.org/; accessed on 17 August 2020). The annotation was performed by the topGO package run in R 4.1.0. GO terms with less than 10 annotated genes, and single gene targets were excluded from the analysis. CTRL—mice fed with standard diet, and their descendants; HFD—mice fed with high-fat diet, and their descendants; HFDt—mice fed with high-fat diet for 60 days that was then replaced by a standard diet, and their descendants.

**Figure 7 biomedicines-10-00909-f007:**
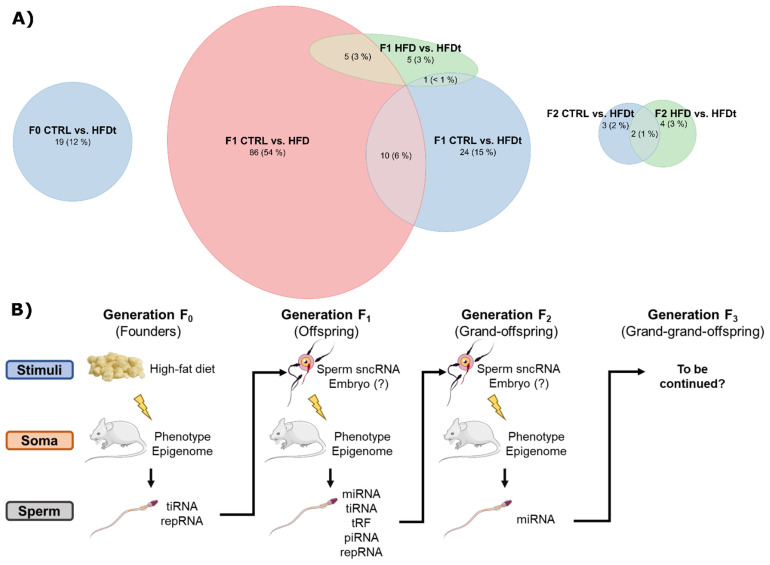
Number of differently expressed sperm sncRNA sequences varies significantly in each generation and in each lineage of ancestral exposure to HFD. (**A**) Euler diagram illustrating the scale of differently expressed sperm sncRNA sequences in every generation and considering all the comparisons: HFD vs. CTRL, HFDt vs. CTRL, and HFDt vs. HFD. The breakdown of these numbers according to sncRNA biotype is provided in Table 1. (**B**) Schematic representation of “inherited metabolic memory” of testicular cells via sperm sncRNA. The exposure of the founder mice to HFD is the stimulus driving epigenetic changes in somatic cells, including Sertoli cells and spermatocytes. These changes are reflected in the sperm of the founders, which presented alterations in the number of tiRNAs and repRNAs, compared to CTRL. This epigenetic fingerprint will be the stimulus for epigenomic change in the offspring, especially during embryo development. Thus, the soma of the offspring will have a different epigenetic configuration that will, in its turn, influence the epigenome and the sncRNA content of produced sperm, distinct from the ancestors. This sperm sncRNA fingerprint will be the stimulus for the next generation, influencing the embryo development and the epigenome of the grand-offspring of the exposed mice.

**Table 1 biomedicines-10-00909-t001:** Number of differently expressed sncRNA sequences grouped by biotype. The detailed annotation of the differently expressed sequences, sorted by biotype, is provided in Appendix A. Abbreviations: miRNA—micro RNA; tsRNA—tRNA-derived small RNA; tRF—tRNA-derived fragment; tiRNA—transcription initiation RNA; piRNA—piwi-interacting RNA; repRNA—repeat-derived small RNA.

			tsRNA			
Generation	Comparison	miRNA	tRF	tiRNA	piRNA	repRNA
	CTRL vs. HFD	0	0	0	0	0
**F0**	CTRL vs. HFDt	0	0	10	0	9
	HFD vs. HFDt	0	0	0	0	0
	CTRL vs. HFD	2	0	1	18	80
**F1**	CTRL vs. HFDt	0	13	0	1	21
	HFD vs. HFDt	1	4	2	3	1
	CTRL vs. HFD	0	0	0	0	0
**F2**	CTRL vs. HFDt	5	0	0	0	0
	HFD vs. HFDt	6	0	0	0	0

## Data Availability

The RNA-seq data and the scripts generated in this study are provided as source data (https://doi.org/10.5281/zenodo.6001903; accessed on 8 March 2022). All other data are available from the corresponding author upon reasonable request.

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
