# Peer review of "Testicular “Inherited Metabolic Memory” of Ancestral High-Fat Diet Is Associated with Sperm sncRNA Content"

_biomedicines, 2022, doi:10.3390/biomedicines10040909_

Round 1

Reviewer 1 Report

Manuscript Number:

In the manuscript entitled :“ Testicular inherited metabolic memory”  of ancestral high-fat  diet is associated with sperm sncRNA content” Luís Chrisostomo and colleagues analyzed sperm RNA by sequencing in genotypes of C57BL/6 mice under two conditions on a high-fat diet (HFD) to find a heritability factor (RNA) of obesity and diabetes phenotypes observed in the F1 and F2 generations.
The author conclude “Our results suggest that sperm sncRNA content is influenced by ancestral exposure to HFD, contributing to the sperm epigenome up to the grand-offspring”.
However, it remains unclear that sncRNAs act as signaling factors under HFD.

In Figure 1, the CTRL (F0, F1 and F2) show the same animals under the same diet, and variations are visible from F0 to F1 and to F2 between the same animals under the same conditions. What is the explanation, and in fact, not even commented?  Does it suggest that the variations detected here are independent of diet? In each group seems an oscillation occurs from one generation to another, when one signal is down in F0, then goes up in F1 etc.

Figure 7 is interesting and should help to have an overall view, but it will be more informative to compare F0 CTRL to F1 and to F2 and the same for each of the groups HFD and HFDt. If the results are consistent then, in addition, at least candidates sncRNAs would have to be validated by Q-RT-PCR in individual samples. 
As for the type of variations observed, it will be more judicious to compare in the same family sperm of the father to grandson F1 and F2.
As the reported variations are very large in animals with the same genotypes it is difficult to extend the result to humans with their fully heterogenous genotypes.

The data are simply not strong enough for the moment for the conclusions to be extended to humans with their fully heterogenous genotypes.

Author Response

We thank the reviewer for the insightful comments on our manuscript. The limitations highlighted by the reviewer are fair and were subjected to extensive discussion among the authors while preparing the manuscript. We have done our best efforts to address the comments, bringing this important discussion into the paper. Thank you.

Comments to be addressed:

  1. “In the manuscript entitled: “Testicular inherited metabolic memory” of ancestral high-fat diet is associated with sperm sncRNA content” Luís Crisóstomo and colleagues analyzed sperm RNA by sequencing in genotypes of C57BL/6 mice under two conditions on a high-fat diet (HFD) to find a heritability factor (RNA) of obesity and diabetes phenotypes observed in the F1 and F2 generations. The authors conclude “Our results suggest that sperm sncRNA content is influenced by ancestral exposure to HFD, contributing to the sperm epigenome up to the grand-offspring”. However, it remains unclear that sncRNAs act as signaling factors under HFD.”

We thank the reviewer for this important comment. We agree with the reviewer’s remark that, in this work, it remains unclear whether sncRNAs are signaling factors of HFD. In fact, we were cautious to not overstate our observations, and therefore we just report an association, and not causation, between ancestral HFD and sperm epigenome. We have now stressed this limitation in the abstract: “Although the causality between sperm sncRNAs content and transgenerational epigenetic inheritance of HFD-related traits remains elusive, our results suggest that sperm sncRNA content is influenced by ancestral exposure to HFD, contributing to the sperm epigenome up to the grand-offspring.” Thank you.

  1. In Figure 1, the CTRL (F0, F1 and F2) show the same animals under the same diet, and variations are visible from F0 to F1 and to F2 between the same animals under the same conditions. What is the explanation, and in fact, not even commented? Does it suggest that the variations detected here are independent of diet? In each group seems an oscillation occurs from one generation to another, when one signal is down in F0, then goes up in F1 etc.

Thanks for the comment. We stress that the animals are different in every generation, although they are part of the same family, i.e. F1 CTRL are the offspring of CTRL mice, and F2 CTRL are the grand-offspring of CTRL mice. One of the biggest challenges of multigenerational studies is ensuring that all animals are reared in the exact same conditions. Despite the best efforts, there are always oscillations caused by uncontrolled variables, such as seasonal and environmental variables that may change the alertness and stress status of the animals. Therefore, in order to limit the effect of intergenerational variance, experimental designs based on transgenerational models exclusively compare animals which have lived during the same period. In fact, when comparing control animals from different generations, we have found differences in tiRNAs and tRFs between F0 and F2, in piRNAs and repRNAs between F0 and F1, but no differences in miRNAs. Therefore, we have adopted the same strategy as other studies with similar animal models (1, 2) and omitted the intergenerational results, as we consider that no reliable comparisons can be drawn from those. We have now included a note in the “Results” section: “We have only compared animals which have lived during the same period, to limit the influence of uncontrollable variables that may affect sperm sncRNA content, such as seasonal and environmental variation. This analytical strategy has been commonly adopted in transgenerational inheritance models (1, 2) Thank you.

  1. “Figure 7 is interesting and should help to have an overall view, but it will be more informative to compare F0 CTRL to F1 and to F2 and the same for each of the groups HFD and HFDt. If the results are consistent then, in addition, at least candidates sncRNAs would have to be validated by Q-RT-PCR in individual samples.”

We thank the reviewer for this comment. We agree that Q-RT-PCR could be used to validate the expression of sncRNA candidates found using Next Generation RNAseq, despite the differences between the techniques. However, it is not possible to undergo such validation in due time to make it worth. Such validation requires a satisfactory number of samples, custom reagents and protocol optimization that could be included as an independent paper. We have further discussed the need for further validation of our observations by Q-RT-PCR: “Additionally, qRT-PCR could be used to validate our sncRNA candidates, increasing the number of biological replicates. Although NGS and qRT-PCR are distinct techniques, their results correlated fairly well, even for miRNA expression (3).” Thank you.

  1. “As for the type of variations observed, it will be more judicious to compare in the same family sperm of the father to grandson F1 and F2”.

Thanks for the comment. As we have previously discussed, it is expected some oscillation among animals from the same family from one generation to the next, due to uncontrollable environmental variables.  Thank you.

  1. “As the reported variations are very large in animals with the same genotypes it is difficult to extend the result to humans with their fully heterogenous genotypes. The data are simply not strong enough for the moment for the conclusions to be extended to humans with their fully heterogenous genotypes.”

Thanks for the comment. We absolutely agree with the reviewer’s remark. Once again, we have tried to be cautious to not overstate the reach of our observations. It has already been reported in humans a transgenerational association between food availability and cardiovascular disease (4), and also that sperm sncRNA content is remodeled by adiposity and diet (5, 6). However, we are aware that our findings cannot be simply translated to human health, and further studies are needed. Regarding that, we have highlighted this limitation in our conclusions: “Yet, further studies are needed to translate our findings to human health. Now, we suggest that sperm sncRNAs are a potentially meaningful marker of metabolic and functional changes in testes due to ancestral exposure to HFD.” Thank you.

Bibliography:

  1. Gapp K, Jawaid A, Sarkies P, Bohacek J, Pelczar P, Prados J, et al. Implication of sperm RNAs in transgenerational inheritance of the effects of early trauma in mice. Nature neuroscience. 2014;17(5):667-9. 10.1038/nn.3695
  2. Carone BR, Fauquier L, Habib N, Shea JM, Hart CE, Li R, et al. Paternally induced transgenerational environmental reprogramming of metabolic gene expression in mammals. Cell. 2010;143(7):1084-96. 10.1016/j.cell.2010.12.008
  3. Git A, Dvinge H, Salmon-Divon M, Osborne M, Kutter C, Hadfield J, et al. Systematic comparison of microarray profiling, real-time PCR, and next-generation sequencing technologies for measuring differential microRNA expression. RNA. 2010;16(5):991-1006. 10.1261/rna.1947110
  4. Kaati G, Bygren LO, Edvinsson S. Cardiovascular and diabetes mortality determined by nutrition during parents' and grandparents' slow growth period. European Journal of Human Genetics. 2002;10(11):682-8. 10.1038/sj.ejhg.5200859
  5. Donkin I, Versteyhe S, Ingerslev Lars R, Qian K, Mechta M, Nordkap L, et al. Obesity and Bariatric Surgery Drive Epigenetic Variation of Spermatozoa in Humans. Cell Metab. 2016;23(2):369-78. 10.1016/j.cmet.2015.11.004
  6. Nätt D, Kugelberg U, Casas E, Nedstrand E, Zalavary S, Henriksson P, et al. Human sperm displays rapid responses to diet. PloS Biol. 2019;17(12). 10.1371/journal.pbio.3000559

Reviewer 2 Report

I found this an excellent and very thought-provoking article. I am still insecure about just what statistically significant changes in sncRNA expression means in real or physiological terms, as I suspect are the authors. One reads an implicit need to find some kind of 'causality', but this I think must remain elusive. Although I cannot fault the statistics which have been applied, I do wonder whether, if (as is not possible) the whole study would be repeated (using genetically - and ?epigenetically - unrelated, non-sibling mice, or even other species, as founders), the same individual differentially expressed sncRNAs would be highlighted. I suspect not, but possibly classes would be. Or possibly, we are seeing here merely a molecular 'fingerprint' reflecting a historical event which has predictable repercussions through the generations. I have no specific comments or corrections to suggest, rather the manuscript should be given the opportunity of exposure to the scientific community as it is.

Author Response

Reviewer #2

We thank the reviewer for the constructive comments. As the reviewer states, the findings reported in this paper are quite provocative, especially considering our previous observations (1, 2), but many questions remain unanswered. Thank you.

Comments to be addressed:

  1. “I found this an excellent and very thought-provoking article. I am still insecure about just what statistically significant changes in sncRNA expression means in real or physiological terms, as I suspect are the authors. One reads an implicit need to find some kind of 'causality', but this I think must remain elusive.”

We thank the reviewer for the comment. We agree with the reviewer on the elusiveness of a causal relationship between HFD, sperm sncRNA content and transgenerational health outcomes. As such, we have put an extraordinary effort to not overstate the implications of our findings, particularly regarding the molecular mechanisms underlying this potential causality. We have now further highlighted this limitation in the Abstract: “Although the causality between sperm sncRNAs content and transgenerational epigenetic inheritance of HFD-related traits remains elusive, our results suggest that sperm sncRNA content is influenced by ancestral exposure to HFD, contributing to the sperm epigenome up to the grand-offspring.” Thank you.

  1. “Although I cannot fault the statistics which have been applied, I do wonder whether, if (as is not possible) the whole study would be repeated (using genetically - and ?epigenetically - unrelated, non-sibling mice, or even other species, as founders), the same individual differentially expressed sncRNAs would be highlighted. I suspect not, but possibly classes would be. Or possibly, we are seeing here merely a molecular 'fingerprint' reflecting a historical event which has predictable repercussions through the generations.”

We thank the reviewer for this suggestion. We hope that our results may lead to new studies to clarify the mechanisms underlying our observations. Future studies could clarify whether HFD or adiposity is the main factor for the changes observed in sperm sncRNA content. Besides, mechanistic studies are needed to clarify the impact of this sperm sncRNA fingerprint in embryo development. Some studies have already studied the role of specific sperm sncRNAs in the transmission of acquired traits (3), but it is also worthy to study other epigenetic mechanisms as DNA methylation and histone retention. We have included new suggestions for future work: “However, to support this inheritance mechanism, future studies must investigate other epigenetic factors carried by sperm in response to HFD and HFD-related adiposity, as well as the impact of HFD and adiposity on the epigenome of germ cells and testicular cells, notably Sertoli cells. Besides, the impact of altered sperm sncRNA content in embryo development could further studied, for instance by injecting sperm sncRNAs isolated from HFD-exposed individuals into naïve zygotes (3).” Thank you.

Bibliography:

  1. Crisóstomo L, Jarak I, Rato LP, Raposo JF, Batterham RL, Oliveira PF, et al. Inheritable testicular metabolic memory of high-fat diet causes transgenerational sperm defects in mice. Sci Rep. 2021;11(1):9444. 10.1038/s41598-021-88981-3
  2. Crisóstomo L, Videira RA, Jarak I, Starčević K, Mašek T, Rato L, et al. Inherited metabolic memory of high-fat diet impairs testicular fatty acid content and sperm parameters. Molecular Nutrition and Food Research. 2022;66(5):2100680. 10.1002/mnfr.202100680
  3. Chen Q, Yan M, Cao Z, Li X, Zhang Y, Shi J, et al. Sperm tsRNAs contribute to intergenerational inheritance of an acquired metabolic disorder. Science. 2016;351(6271):397-400. 10.1126/science.aad7977

Round 2

Reviewer 1 Report

The author, have better defined their goals.

However, I don't still understand this sentence:

 “Although, NGS and qRT-PCR are distinct techniques, their results correlated fairly well, even for miRNA expression”

I disagree, the results should be reproducible regardless of the technique used. Unlike NGS, RT-PCR is more sensitive technique.  The NGS technique can often miss the signal due to several steps before sequencing.

In Table 1, the name of the miRNAs could be listed and validated by qRT-PCR.

Generation

Comparison

miRNA

F0

CTRL vs. HFD

2

F1

HFD vs. HFDt

1

F2

CTRL vs. HFDt

5

HFD vs. HFDt

6

Author Response

Comments to be addressed:

  1. However, I don't still understand this sentence: “Although, NGS and qRT-PCR are distinct techniques, their results correlated fairly well, even for miRNA expression.” I disagree, the results should be reproducible regardless of the technique used. Unlike NGS, RT-PCR is more sensitive technique. The NGS technique can often miss the signal due to several steps before sequencing.

We are glad to discuss this aspect with the reviewer. We would like to clarify that we have stated that the results of NGS and qRT-PCR are comparable. However, there are differences between the techniques, as the reviewer points out. We agree that the extra pre-processing steps of NGS may exclude sequences with low expression, but it is unlikely that those sequences will have a biological relevant role. Also, NGS does not have the same dynamic range limitations as precursor technologies, such as microarrays, and its sensitivity is comparable to qRT-PCR (1). We have adapted the previous claim to make it clearer and to highlight the differences between the two methods: “Additionally, qRT-PCR could be used to validate our sncRNA candidates, increasing the number of biological replicates. Although NGS and qRT-PCR are distinct techniques, especially regarding normalization and statistical analyses, it has been demonstrated that the results are comparable, even for miRNA expression (2).” Thank you.

  1. In Table 1, the name of the miRNAs could be listed and validated by qRT-PCR.

We thank the reviewer for this suggestion. We decided to keep Table 1 to preserve the coherence of the information provided by it. However, we acknowledge the relevance of the information requested by the reviewer and, as an alternative, we have compiled a new Supplementary Table 1, containing all the differently expressed sequences, sorted by sncRNA biotype. Thank you.

Bibliography:

  1. Wang Z, Gerstein M, Snyder M. RNA-Seq: a revolutionary tool for transcriptomics. Nature Reviews Genetics. 2009;10(1):57-63. 10.1038/nrg2484
  2. Git A, Dvinge H, Salmon-Divon M, Osborne M, Kutter C, Hadfield J, et al. Systematic comparison of microarray profiling, real-time PCR, and next-generation sequencing technologies for measuring differential microRNA expression. RNA. 2010;16(5):991-1006. 10.1261/rna.1947110